# A Rapid and Specific Genotyping Platform for *Plasmodium falciparum* Chloroquine Resistance via Allele-Specific PCR with a Lateral Flow Assay

Weijia Cheng,[a] Xiaonan Song,[a] Huiyin Zhu,[a] Kai Wu,[b] Wei Wang,[c] Jian Li[a]

[a]School of Basic Medical Sciences, Hubei University of Medicine, Shiyan, China
[b]Department of Schistosomiasis and Endemic Diseases, Wuhan City Center for Disease Prevention and Control, Wuhan, China
[c]Jiangsu Institute of Parasitic Diseases, Wuxi, China

Weijia Cheng and Xiaonan Song contributed equally to this article. The author order was determined randomly.

**ABSTRACT** Single-nucleotide polymorphisms and genotyping related to genetic detection are several of the focuses of contemporary biotechnology development. Traditional methods are complex, take a long time, and rely on expensive instruments. Therefore, there is an urgent need for a rapid, simple, and accurate method convenient for use in resource-poor areas. Thus, a platform based on allele-specific PCR (AS-PCR) combined with a lateral flow assay (LFA) was developed, optimized, and used to detect the genotype of the *Plasmodium falciparum* chloroquine transporter gene (*pfcrt*). Subsequently, the system was assessed by clinical isolates and compared with Sanger sequencing. The sensitivity and specificity of the AS-PCR-LFA platform were 95.83% (115/120) and 100% (120/120), respectively, based on the clinical isolates. The detection limit of plasmid DNA was approximately $3.38 \times 10^5$ copies/$\mu$L. In addition, 100 parasites/$\mu$L were used for the dried filter blood spots from clinical isolates. The established rapid genotyping technique is not limited to antimalarial drug resistance genes but can also be applied to genetic diseases and other infectious diseases. Thus, it has realized the leap and transformation from scientific research theory to practical application and actively responds to the point-of-care testing policy.

**IMPORTANCE** Accurate recognition of the mutation and genotype of genes are essential for the treatment of infectious diseases and genetic diseases. Based on the techniques of allele-specific PCR (AS-PCR) and a lateral flow assay (LFA), a rapid and useful platform for mutation detection was developed and assessed with clinical samples. It offers a powerful tool to identify antimalarial drug resistance and can support malaria control and elimination globally.

**KEYWORDS** allele-specific PCR, chloroquine resistance, genotyping, lateral flow assay, *Plasmodium falciparum*

Despite significant progress in malaria control and elimination, malaria remains a major burden of mosquito-borne infectious diseases globally. According to the World Health Organization (WHO) (1), there were 228 million malaria cases and 602,000 deaths worldwide in 2020. Most were centralized in resource-poor countries and areas, particularly in sub-Saharan Africa and Southeast Asia. Chloroquine (CQ) has been the mainstay of the treatment for *Plasmodium falciparum* since the 1940s (2). However, the development of drug resistance reduces the efficacy of CQ (3). The emergence of the triple mutant haplotype, named CV**IET**, in the *P. falciparum* chloroquine transporter gene (*pfcrt*), has been confirmed in several countries to increase the risk of CQ treatment

Address correspondence to Jian Li, yxlijian@163.com.

The authors declare no conflict of interest.

**TABLE 1** Selected and labeled primers for genotype detection in the *pfcrt* gene[a]

| Primer name[b] | Sequence and modification (5′→3′)[c] | Description |
|---|---|---|
| Bio-Pfcrt-CVMNK_Fwd_WT (W') | TATTTAAGTGTATGTGTAATGAAT*A*A | Allele-specific primer |
| Bio-Pfcrt-CV**IET**_Fwd_Mut (M') | TATTTAAGTGTATGTGTAATTGAA*A*C | |
| Dig-Pfcrt_Rev | ATATTGGTAGGTGGAATAGATTCT | Common primer |

[a]Gene identifier PF3D7_0709000.
[b]Bio, biotin; Dig, digoxin; Fwd, forward; Rev, reverse; WT, wild-type; Mut, mutant.
[c]The target sites are shown in bold, underlined. *, the location of phosphorothioate modification.

failure (4–6). Therefore, close and efficient detection of this single-nucleotide polymorphism (SNPs) loci and alleles will benefit medication guidance.

To evaluate the extent of drug resistance with genes, SNPs detection technology is considered one of the main fields conquered by contemporary biotechnology (7). For the detection of *Plasmodium* parasites mutation, nested PCR with sequencing considers the 'gold standard' method for detecting SNPs and genotyping (6, 8, 9). However, due to the long test period and complicated steps, test results cannot be obtained in real-time. More importantly, it brings substantial economic expenditure to resource-deficient regions. For other technologies, similarly multistep procedures, higher costs, and the need for sophisticated equipment limit their application (10–14). At present, the rapid diagnosis of genotypes of *P. falciparum* drug resistance genes is rare. In particular, the use of rapid genotyping methods instead of sequencing in the *pfcrt* gene has also not been reported. The crucial issue is how to analyze and detect disease-related known genotypes quickly and reliably. Additionally, allele-specific PCR (AS-PCR) has firmly established its place in genotyping detection (15). However, the significant weakness of this method is the high number of false-positives in the analysis (16). The lateral flow assay (LFA) (17–19) has been suggested as an effective alternative to Sanger sequencing. Sequencing results often take days and require additional expensive testing equipment. LFA detection can be initiated by rapid nucleic acid diagnosis strip contacts with the PCR product. It has the advantage that it does not need additional complicated equipment and manipulations.

Here, a detection platform named AS-PCR-LFA based on AS-PCR combined with LFA was established and assessed with the *pfcrt* gene as the target. On this basis, the developed diagnosis system was fully optimized. It can effectively screen genotypes, save time and significantly, and offer a reliable scientific basis for personalized medicine. This platform can be applied to other antimalarial drug resistance genes as well as to genetic diseases and infectious diseases.

## RESULTS

**Principle of the AS-PCR-LFA platform.** To detect the sensitivity and specificity of *pfcrt* alleles, an opportunistic AS-PCR-LFA detection system was developed (Fig. 1). After amplification (Table 1), the PCR products can be directly added to the LFA and interpreted according to whether the test line (T line) was colored. Fig. 1A demonstrates the principle of AS-PCR in distinguishing the *pfcrt* alleles. Corresponding allele-specific primers were designed for genotypes. Each specific primer introduced a 3′ terminal double phosphorothioate modification. In the LFA (Fig. 1B), biotin-labeled on the 5′-end of allele-specific primers in PCR products was identified and captured by SA-AuNPs on the conjugate pad and continued to flow forward. When the target products were present, they containing digoxin labeled on the 5′-end of common primers were bound and aggregated by an immobilized anti-digoxin monoclonal antibody that was fixed on the T line. Because of the presence of AS-AuNPs, the T line first presents a gold-red band. The remaining SA-AuNPs continue forward and are then captured by biotin-BSA on the control line (C line), presenting another red band. If there were no target products, there was no red band on the test line. Fig. 1C illustrates that gDNA was extracted from dried filter blood spots (DBS) and then amplified via AS-PCR. The wild-type tube adds the wild-type template and primer, and the mutant-type tube adds the mutant-type template and primer. The PCR will be blocked when the wild-type tube includes the mutant-type

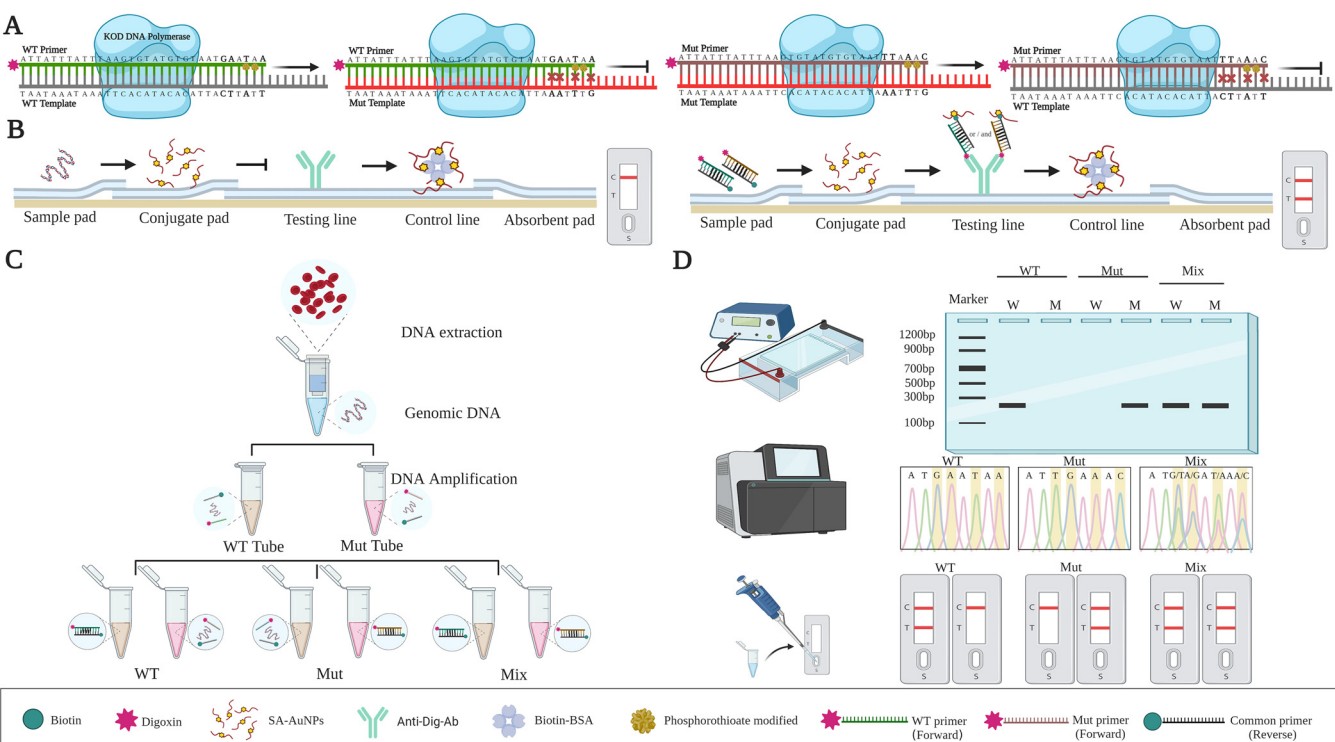

**FIG 1** Schematic illustration of allele-specific PCR (AS-PCR) combined with a lateral flow assay (LFA) system. (A) Principle of AS-PCR for genotype. (B) Structure of labeled lateral flow device. (C) Sample preparation and target amplification. (D) Results were analyzed based on agarose gel electrophoresis, DNA sequencing, and the signal read-out by visual interpretation in LFA. W and M on the sample represent wild-type and mutant-type primers, respectively. C and T represent the control line and test line, respectively.

template and wild-type primer, and the mutant-type tube includes the wild-type template and mutant-type primer. PCR products amplified by AS-PCR can be verified by electrophoresis, sequencing, and the LFA for genotyping (Fig. 1D). Genotyping results can be visually interpreted by the LFA in less than 10 min and without additional equipment and replacing traditional DNA sequencing.

**Plasmids.** The two target sequences were cloned into the vector *pUC57* to generate recombinant plasmids. The plasmids *pUC57*/Pfcrt$_{CVMNK}$ and *pUC57*/Pfcrt$_{CVIET}$ were obtained. Subsequently, the recombinant plasmid was double digested with BamHI and XhoI (Fig. S1A). The expected fragments were detected, with lengths of 3357 bp and 686 bp, respectively (Fig. S1A). Finally, the plasmids were sequenced (Fig. S1B) and were identical to the matched target sequence, indicating that they were successfully constructed.

**Optimization of the AS-PCR-LFA platform.** AS-PCR-LFA was performed at different annealing temperatures: 55, 55.2, 55.5, 56.0, 56.6, 57.2, 57.7, 58.4, 59.0, 59.5, 59.8, and 60.0°C (Fig. 2A). For wild-type primers or mutant-type primers, a single band of the expected size (222 bp) was observed. For CV**IET**, the amplification effect began to weaken gradually when the temperature increased to 58.4°C. Therefore, the most effective annealing temperature of CV**IET** was 56.6°C. For CVMNK, 56.6°C was selected as the appropriate annealing temperature for the convenience of subsequent amplification.

For MgSO₄, different concentrations, including 0, 0.5, 1.0, 1.5, 2.0, 2.5, 3.0 and 3.5 mM, were tested. For CVMNK (Fig. 2B), bands could not be detected without MgSO₄. As the concentration increased, the bands were bright and single at 1.5 mM, while the amplification effect began to weaken gradually when 2.0 mM was added, and nonspecific amplification appeared. For CV**IET**, no bands were amplified at 0.5 mM. Nonspecific amplification began at 2.0 mM. Therefore, at 1.5 mM (Fig. 2B), the targeted band appeared as a single band, and the optimal concentration of MgSO₄ was selected for both CVMNK and CV**IET**.

The primer concentration of each genotype was optimized, and the primer concentrations were assessed at 0.04, 0.1, 0.2, 0.3, 0.4, 0.5, 0.6, and 0.7 μM (Fig. 2C). After optimization,

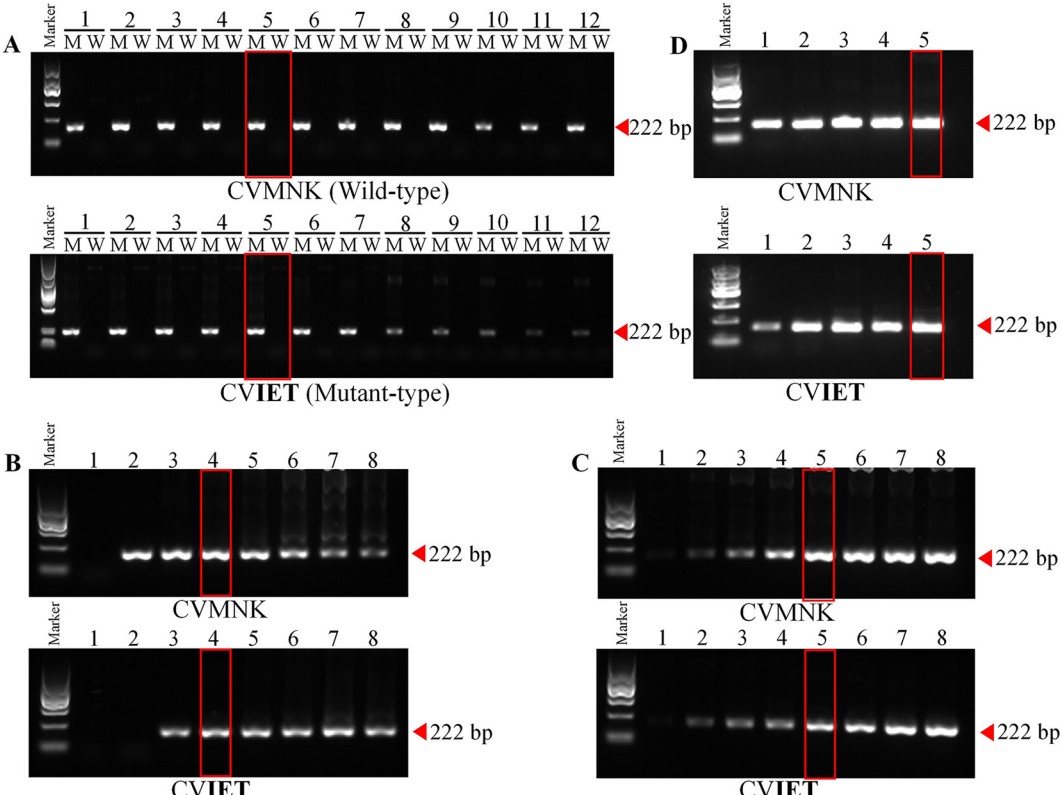

**FIG 2** Optimization of the AS-PCR-LFA detection system. (A) Optimization of the annealing temperature. Lanes 1 to 12 result from annealing temperatures of 55, 55.2, 55.5, 56.0, 56.6, 57.2, 57.7, 58.4, 59.0, 59.5, 59.8, and 60.0°C, respectively. (B) Optimization of MgSO$_4$.1 to 8 represent 0, 0.5, 1.0, 1.5, 2.0, 2.5, 3.0, and 3.5 mM, respectively. (C) Optimization of primer concentration. The numbers 1 to 8 represent final concentrations of 0.04, 0.1, 0.2, 0.3, 0.4, 0.5, 0.6, and 0.7 $\mu$M. (D) Optimization of the number of PCR cycles. The numbers 1 to 5 represent 15, 20, 25, 30, and 35 cycles, respectively. The marker indicates the DNA molecular marker, including 100 bp, 300 bp, 500 bp, 700 bp, 900 bp, and 1200 bp. The brightest band was 700 bp with a mass concentration of 83 ng and the rest of the bands were 42 ng.

there was no nonspecific amplification in the wild-type and mutant-type strains. With increasing primer concentration, the target band gradually strengthened. For CVMNK, a bright and single band could be found when the primer concentration was added to 0.3 $\mu$M. Similarly, for CV**IET**, the specific band could be detected when the primer concentration was added to 0.4 $\mu$M. Overall, the primer concentration at 0.4 $\mu$M for both CVMNK and CV**IET** was confirmed as the final concentration in subsequent experiments (Fig. 2C).

According to the above optimization conditions, the number of cycles was optimized. Single bands were found after 15, 20, 25, 30, and 35 cycles (Fig. 2D). Therefore, 35 cycles were adopted as the optimal cycle number for AS-PCR (Fig. 2D).

**AS-PCR-LFA sensitivity and specificity validation.** Based on the optimized reaction system and conditions, the sensitivity and specificity of AS-PCR were assessed. The final concentration of plasmids was set from $3.38 \times 10^9$ to $3.38 \times 10^1$ copies/$\mu$L (Fig. 3). The optimized PCR system was used. A 25 $\mu$L final volume contained 1.0 $\mu$L of 10 $\mu$M common antisense primer, 2.5 $\mu$L 10 × KOD buffer, 0.5 $\mu$L of 0.5 U of KOD-Plus-Neo, 2.5 $\mu$L of 2 mM dNTPs, 1.5 $\mu$L of 25 mM MgSO4, 1.0 $\mu$L of 10 $\mu$M allele-specific primer, and 1.0 $\mu$L of plasmid DNA (pDNA). The samples were then incubated at 94°C for 3 min, 94°C for 30 sec, 56.6°C for 30 sec, 65°C for 60 sec, for 35 amplification cycles, and 65°C for 5 min. The limit of detection (LOD) of CVMNK in agarose gel electrophoresis, and the LFA was $3.38 \times 10^5$ copies/$\mu$L (Fig. 3A and B). The CV**IET** in electrophoresis was $3.38 \times 10^5$ copies/$\mu$L, and the LFA was $3.38 \times 10^6$ copies/$\mu$L (Fig. 3A and B). For specificity, wild-type allele-specific primers amplified wild-type allele templates, and mutant-type allele templates were not amplified (Fig. 3). The results of mutant-type

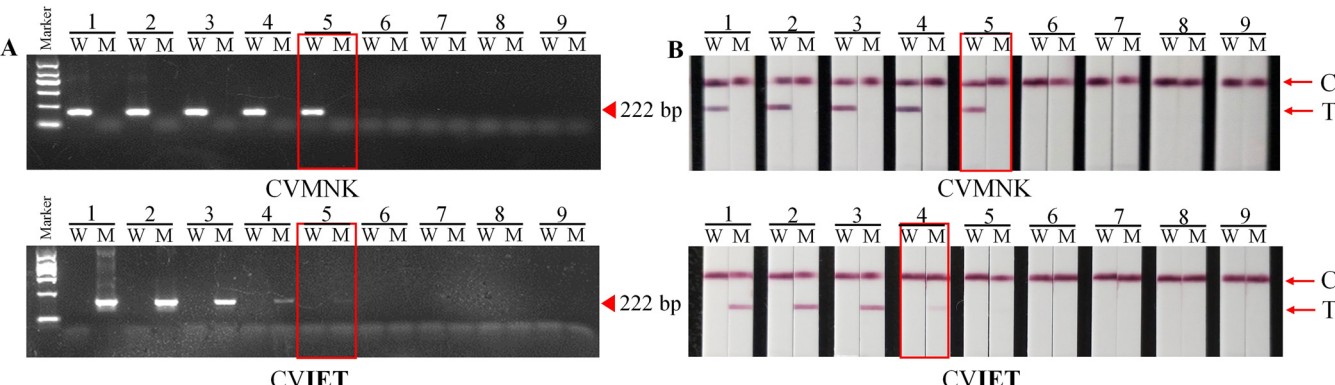

**FIG 3** AS-PCR-LFA sensitivity and specificity validation. (A) Different concentrations of plasmid by agarose gel electrophoresis and the LFA for CVMNK. (B) Different concentrations of plasmid by agarose gel electrophoresis and the LFA for CV**IET**. Lanes 1 to 12 represent the concentrations of wild-type and mutant-type plasmids, which were ranged from $3.38 \times 10^9$ copies/$\mu$L to $3.38 \times 10^1$ copies/$\mu$L, respectively. The marker indicates the DNA molecular marker, including 100 bp, 300 bp, 500 bp, 700 bp, 900 bp, and 1200 bp. The brightest band was 700 bp with a mass concentration of 83 ng and the rest of the bands were 42 ng. C and T represent the control line and test line, respectively.

allele-specific primers were the same, and the amplification band was single without crossover reaction (Fig. 3). The results demonstrated that the established platform had 100% specificity for *pfcrt* genotyping. For LFA, the band signal was consistent with the agarose gel electrophoresis results (Fig. 3). Plasmids were mixed at known concentrations to simulate infections involving two different genotypes. After mixing the plasmids in different proportions, the bands of the wild-type became brighter with the increase in the proportion of plasmids (Fig. S2) and were not disturbed by the mutant plasmids. Similar results were obtained for the mutant-type plasmids.

**Clinical isolate assessment.** To test the dependability of the platform, the certainty was further confirmed with clinical isolates. The ultimate genotyping result of each allele was visually explained by the presence or absence of the color of the T line on the LFA. According to the results, wild isolates of the *pfcrt* gene were successfully amplified with wild-type allele-specific primers, and the target fragment was obtained by electrophoresis (Fig. 4A). The T line was obtained on the wild-type LFA (Fig. 4B). Similarly, mutant allele-specific primers successfully amplified only mutant isolates of the *pfcrt* gene (Fig. 4A), and the T line also displayed color only on mutant-type LFA (Fig. 4B). Two bands were obtained from the amplified mixed-type by electrophoresis (Fig. 4A). Similarly, the T line was colored on both wild-type and mutant LFA (Fig. 4B). Finally, the electrophoresis and LFA results were compared with sequencing results.

The clinical samples were genotyped by Sanger sequencing. Subsequently, AS-PCR-LFA system detection was performed, and the results are shown in Table S1. The methodological evaluation and analysis of the AS-PCR-LFA results and nested PCR with sequencing results were presented in Table 2. The sensitivity, specificity, false-negative rate, and false-positive rate of AS-PCR-LFA for the *pfcrt* gene were 95.83%, 100%, 4.17%, and 0, respectively. For these clinical samples, the sensitivities of CVMNK, CV**IET**, and CV M/**I** N/**E** K/**T** were 96.88% (62/64), 97.06% (32/34), and 90.91% (20/22), respectively (Table 2). For specificity, all of them were 100%. The false-negative rates were 3.12%, 2.94%, and 9.09%, respectively. Finally, the LOD of AS AS-PCR-LFA was 100 parasites/$\mu$L of DBS in 120 clinical samples.

## DISCUSSION

Recently, the antimalarial drug resistance of *P. falciparum* has remained a priority (6, 9) and will hamper the global plan for malaria elimination. Therefore, it is necessary to timely monitor the molecular markers involved in drug-resistant Plasmodium and establish an economical, rapid, and accurate detection method. Although Sanger sequencing is considered the gold standard for genotyping, it is costly and time-consuming (9). Thus, compared with the sequencing method, the method established in the current study is a rapid detection platform for genotyping.

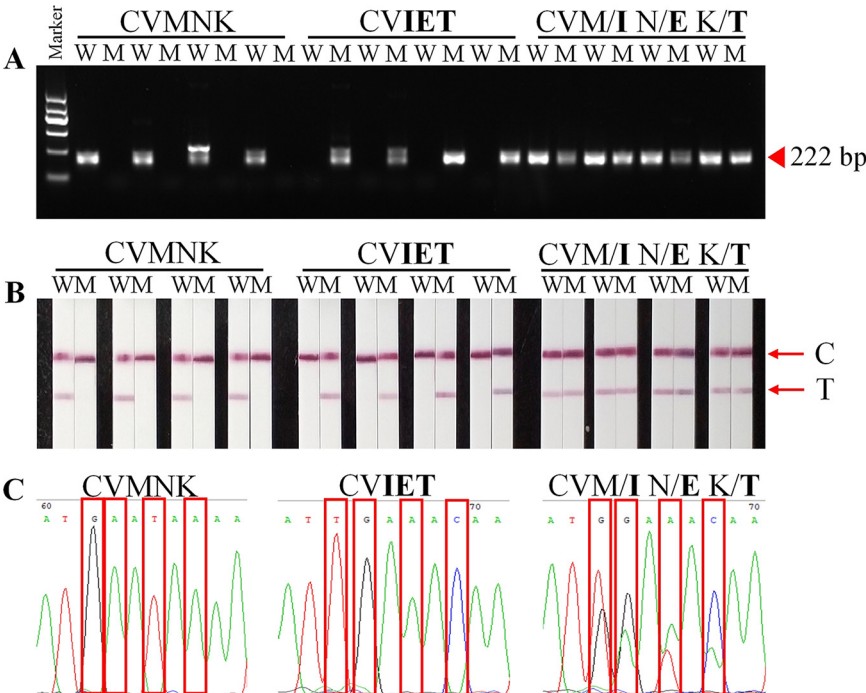

**FIG 4** Genotyping result (taking partial samples as an example). (A) Agarose gel electrophoresis. (B) The AS-PCR-LFA system through visualized interpretation. (C) DNA sequencing. The marker indicates the DNA molecular marker, including 100 bp, 300 bp, 500 bp, 700 bp, 900 bp, and 1200 bp. The brightest band was 700 bp with a mass concentration of 83 ng and the rest of the bands were 42 ng. C and T represent the control line and test line, respectively.

Detection through nested PCR with sequencing is the most used genotyping technique (6, 8, 9). It is characterized primarily by its high specificity, but the sequencing still requires additional equipment, and the long waiting time is still an open problem. Similarly, *P. falciparum* patients may miss the optimal treatment time due to long waiting for the results of an evaluation of drug resistance. Real-time fluorescence quantitative PCR (qPCR) can also be used for genotyping of *pfcrt* genes (20) without agarose gel electrophoresis and sequencing to interpret the results. However, there is also the problem of relying on laboratory instruments, which cannot be carried out in the field, and it is not easy to promote in resource-poor areas, particularly in the WHO African region. A recent study demonstrated that genotyping can also be performed via DNA microarray using specific probes attached to microarrays (11), but the method still required expensive equipment and specialized operators to perform it. The AS-PCR technique, as a simple, effective, and low-cost genotyping technique has been used multiple in the detection of *P. falciparum* resistance genes in recent years (21), and withal, it has the problem of false positives and still needs to be verified by agarose gel electrophoresis and sequencing. Loop-mediated isothermal amplification (LAMP) (22) is also generally well-received in genotyping because it can be easily amplified at a

**TABLE 2** Methodological comparison of AS-PCR-LFA and Nested PCR with sequencing

| | Method | | Sensitivity (no. %) | Specificity (no. %) | False-negative (no. %) | False-positive (no. %) |
|---|---|---|---|---|---|---|
| Genotype | Nested PCR with sequencing (no.)[a] | AS-PCR-LFA | | | | |
| CVMNK | 64 | 62 | 96.88 | 100 | 3.12 | 0.00 |
| CV**IET** | 34 | 33 | 97.06 | 100 | 2.94 | 0.00 |
| CV M/**I** N/**E** K/**T** | 22 | 20 | 90.91 | 100 | 9.09 | 0.00 |
| Total | 120 | 115 | 95.83 | 100 | 4.17 | 0.00 |

[a]No. is the number of isolates.

single temperature. However, its primer design is difficult, and its false-positive rate is high, which makes it difficult to popularize widely. The need for attention to sensitivity, specificity, cost, testing time, and throughput is important when developing a method for widespread use in resource-poor areas. In 2017, LAMP combined with the LFA method was established to detect the N51**I** mutation site of the *pfdhfr* gene (23). However, the LOD was only 2 ng/$\mu$L. Subsequently, the AS-PCR-LFA rapid detection technique was established and optimized (21) in our previous study, and the LOD of N51**I** in the *pfdhfr* gene was only 20 fg/$\mu$L. After AS-PCR amplification, low-cost and rapid LFA was used to replace expensive and time-consuming Sanger sequencing. It can be widely used in resource-poor areas that cannot afford sequencing costs. The present study attempted to use the AS-PCR-LFA system to detect the wild-type and mutant-type *pfcrt* genes. Previous studies, including ours, mainly focused on the development of SNP detection. To our knowledge, there are no new protocols for genotyping, and a rapid method is still lacking. Thus, the platform of AS-PCR-LFA for *pfcrt* gene genotyping was developed in the current study. Based on a previous study of SNP detection, this is a positive attempt. This indicates that the application scope and scenarios of AS-PCR-LFA are expanded.

Compared with the previous results, there is no nonspecific amplification after optimization and no need to carry out different amplification systems for different densities of parasitemia, further shortening the detection time. In the optimization of the AS-PCR-LFA system, 56.6°C was selected as the best temperature to avoid the annealing temperature being too high, which would affect the sensitivity of subsequent experiments. In addition, in the selection of the optimal primer concentration, 0.4 $\mu$M was selected to reduce the cost. The *pfcrt* gene as a genotype has four mismatch bases between wild-type and mutant-type, which means that it was not required to introduce additional artificial mismatch to increase the specificity of the allele-specific primers. In addition, the problem of false-positives in AS-PCR technology (15) was solved by using KOD DNA polymerase and phosphorothioate modification (21, 24) and introducing an artificial mismatch at the target site. The results of genotyping can be provided quickly by color changes on T lines on the LFA. At present, the established AS-PCR-LFA system has improved the availability of molecular methods in resource-poor regions such as Africa. *P. falciparum* is a particularly difficult organism to amplify and sequence, with more than 80% AT content in genome regions and introns and many AT-rich repeats (25). Therefore, the amplification of *P. falciparum* clinical samples still requires two-step amplification to increase the success rate of amplification (6, 8, 9). Although pDNA was detectable at $3.38 \times 10^5$ copies/$\mu$L, in clinical samples, the LOD was 100 parasites/$\mu$L of DBS. However, in CV**IET** sensitivity, LFA showed weak bands only visible to the naked eye at $3.38 \times 10^5$ copies/$\mu$L, which may be because LFA added only 2.0 $\mu$L PCR products, while electrophoresis added 5.0 $\mu$L. If the PCR products in the LFA were increased, the results were probably identical to or more than the electrophoretic LOD results. It also reflected that the sensitivity of LFA was overtaking the electrophoretic results. In addition, when there is a mixed infection, the method can accurately differentiate genotypes without interference from another genotype. This indicates that the platform is effectual and precise for allelic genotyping in clinical isolates with various parasitemia. It offers a convenient and fast molecular diagnosis method for the clinical genotyping of *P. falciparum*. In addition, under high purity conditions, the recombinant pDNA did not significantly influence the amplification procedure. However, the content of gDNA obtained from blood samples is complex, and there are a variety of PCR inhibitors, which greatly affect the amplification efficiency. Therefore, the small sample of this study failed to be amplified successfully. Finally, the *pfcrt* drug resistance genotype detection system established and optimized is highly specific and sensitive, which also delivers a potential platform for the application and development of genotypes in other infectious and genetic diseases.

According to the deficiencies in this study, it still needs to be improved. The LOD (100 parasites/$\mu$L) still needs further improvement. In future work, the use of digital

PCR that can detect individual copy numbers will be considered to further increase the LOD. Moreover, based on the point-of-care testing (POCT) strategy, the simplified version of the gDNA extraction method is considered to replace the current protocol. Meanwhile, thermostatic amplification techniques, particularly recombinase polymerase amplification, should be considered, which greatly saves the time of amplification (26) and simplifies the operation. By introducing CRISPR/Cas12a technology, the sensitivity and specificity of the method will be further improved. In addition, the developed method in this study can only detect known genotypes, while for unknown genotypes, advanced single-molecule real-time (SMRT) sequencing technology (27) and nanopore sequencing technology (28) must be considered seriously.

In conclusion, the established rapid genotyping technique via AS-PCR-LFA is not only limited to drug resistance genes of *P. falciparum* but can also be applied to a wide range of other fields. This further promotes the POCT strategy for drug resistance gene mutation detection. This technique can also obtain fast and accurate genotyping results and further bring huge social and economic benefits. It also contributes to the cause of individual medical care and national health.

## MATERIALS AND METHODS

**Chemicals and reagents.** All chemicals were of analytical purity and used without further purification. The kits for gDNA extraction and plasmid extraction were purchased from Tiangen Biotech Co., Ltd. (Beijing, China). Agarose and restriction endonucleases, including BamHI and XhoI, were purchased from Thermo Fisher Scientific Inc. (Waltham, MA, USA). PCR was performed using KOD-Plus-Neo from TOYOBO Co., Ltd. (Shanghai, China), and Phanta Max Master Mix from Nanjing NOVIZAN Tech. Co., Ltd. (Nanjing, China). Streptavidin-immobilized gold nanoparticles (SA-AuNPs) and biotinylated bovine serum albumin (Biotin-BSA) were purchased from Beijing Biosynthesis Biotechnology Co., Ltd. (Beijing, China) and Solarbio (Beijing, China), respectively. The consumables required to test the strip system were obtained from Shanghai Kinbio Tech. Co., Ltd. (Shanghai, China). A T100 Thermal Cycler (Bio-Rad) was used for performing the PCR steps. Sequencing was carried out in ABI 3730XL.

**Plasmid construction and identification.** The sequence of the *pfcrt* gene (PF3D7_0709000) from the *P. falciparum* 3D7 strain was obtained from PlasmoDB (http://plasmodb.org/plasmo/, Release 56, 15 Feb 2022) (29). The truncated fragments (686 bp) of *pfcrt* with different genotypes were synthesized by Genewiz Biotechnology Ltd. (Soochow, China). Subsequently, the fragments were cloned into the vector *pUC57* to generate the recombinant pDNA named *pUC57*/Pfcrt$_{CVMNK}$ (wild-type) and *pUC57*/Pfcrt$_{CVIET}$ (mutant-type). Subsequently, they were validated through restriction digestion and sequencing. Finally, concentration and quality were evaluated by Gene5 (Thermo Fisher Scientific, Wilmington, DE, USA).

**Oligonucleotides.** The rules of allele-specific primer design proposed in the previous work were followed (21), with slight modifications. Briefly, each specific primer utilized 3′ terminal double phosphorothioate modification, and there were four different bases between the wild-type and mutant-type genotypes of *pfcrt* that were not required to introduce additional artificial mismatch. Therefore, two allele-specific forward primers and a universal reverse primer were designed to distinguish the corresponding alleles and evaluated by using Oligo 7 (Table 1).

**Development of the AS-PCR-LFA platform.** The annealing temperature, cycle numbers, concentrations of MgSO$_4$, and primers in the reaction system significantly influence the amplification efficiency. Thus, to improve the sensitivity and specificity of target gene amplification, PCR conditions were optimized. The plasmid was diluted to 1:10 and used as the template for system optimization. The lateral flow strip (LFA) (Fig. 1A) (width, 4 mm) consists of five modules, including a 17 mm sample pad, 7 mm conjugate pad, 25 mm NC membrane, and 18 mm absorbent pad, which were fixed to a viscous PVC panel backing with a 2-mm overlap (21). Of note, sample pads and bonding pads were pretreated in suspension buffers containing 0.05 mol Tris-HCl buffer (pH 8.2), BSA, sucrose, and Triton X-100. Subsequently, these pretreated materials were dried overnight at 56°C. Next, biotin-BSA and anti-Dig Ab were fixed on the C and T lines, respectively, using the XYZ Platform Dispenser (Model: HM3035) (Fig. 1B). In addition, the buffer containing SA-AuNPs was fixed on the binding pad. Then, the samples were dried overnight at 56°C. On the LFA, 2.0 $\mu$L PCR products were dripped onto the sample pad, followed by 60 $\mu$L buffer (pH 7.4, 1 × PBS). The results were determined in approximately 10 min by the presence of red lines (Fig. 1D). Similarly, the genotyping results of the test strip can replace the images of agarose gel electrophoresis and sequencing data.

**Sensitivity and specificity validation of AS-PCR-LFA.** As the template, the plasmids *pUC57*/Pfcrt$_{CVMNK}$ and *pUC57*/Pfcrt$_{CVIET}$ were diluted in different gradients. For AS-PCR, the final working concentrations of pDNA ranged from $3.38 \times 10^1$ to $3.38 \times 10^9$ copies/$\mu$L. The reaction was carried out under optimum conditions. The qualitative and quantitative LODs were defined as the plasmid concentration corresponding to the lightest color visible to the naked eye. The LOD was used as the standard for sensitivity evaluation. Meanwhile, the *pUC57*/Pfcrt$_{CVMNK}$ and *pUC57*/Pfcrt$_{CVIET}$ pDNA were cross-amplified with wild-type and mutant-type allele-specific primers. The specificity was evaluated by observing whether there was a specific amplification strip. In addition, *pUC57*/Pfcrt$_{CVMNK}$ and *pUC57*/Pfcrt$_{CVIET}$ were mixed to simulate the interference of different infections on genotyping results. The initial concentration of the

wild-type and mutant plasmids was $3.38 \times 10^7$ copies/$\mu$L. Multiple infections were a mixture of wild-type and mutant at a ratio of 1:9, 2:8, 3:7, 4:6, 5:5, 6:4, 7:3, 8:2, and 9:1, and the wild-type and mutant-type primers were used to amplify the mixed plasmids. Analysis of the degree of interference with genotyping by different proportions of mixed infections according to the brightness of bands on electrophoresis and the LFA.

**Clinical evaluation.** DBS from patients infected with *P. falciparum* parasites have been prepared in a previous project (6, 9). The gDNA from DBS was extracted using a TIANamp blood spots DNA kit following the manual operation. Genotyping of *pfcrt* was confirmed by nested PCR followed by Sanger sequencing. Then, 120 isolates were randomly selected, including wild-type, mutant-type, and mixed-type *pfcrt*. Two-step PCR was performed for genotype detection, and then the results were read according to the color change of the bands on the LFA and agarose gel electrophoresis. Finally, the results were compared with sequencing results for validation

## SUPPLEMENTAL MATERIAL

Supplemental material is available online only.

**SUPPLEMENTAL FILE 1**, PDF file, 0.6 MB.

**SUPPLEMENTAL FILE 2**, XLS file, 0.04 MB.

## ACKNOWLEDGMENTS

J.L. was supported by the Principal Investigator Program of Hubei University of Medicine (grant number HBMUPI202101) and the National Natural Science Foundation of China (grant number 81802046). The funders had no role in the study design, data collection, interpretation, or the decision to submit the work for publication.

We declare no conflict of interest.

The roles for J.L. included conceptualization, formal analysis, funding acquisition, investigation, methodology, project administration, resources supervision, validation, writing original draft, and writing review & editing. The roles for W.J.C. included data curation, methodology, validation, and writing the original draft. The roles for X.N.S. included formal analysis methodology validation. The role of H.Y.Z. was methodology. The roles for K.W. included methodology and resources. The roles for W.W. included methodology, validation, and writing the original draft.

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
