## [Reviewer comments · Microbiology Spectrum]

Microbiology Spectrum

A rapid and specific genotyping platform for *Plasmodium falciparum* chloroquine resistance via allele-specific PCR with lateral flow assay

Weijia Cheng, Xiaonan Song, Huiyin Zhu, Kai Wu, Wei Wang, and Jian Li

Corresponding Author(s): Jian Li, Hubei University of Medicine

Review Timeline:

Submission Date:	December 22, 2021
Editorial Decision:	February 6, 2022
Revision Received:	March 25, 2022
Accepted:	March 25, 2022

Editor: Tim Downing

Reviewer(s): Disclosure of reviewer identity is with reference to reviewer comments included in decision letter(s). The following individuals involved in review of your submission have agreed to reveal their identity: Claire Kamaliddin (Reviewer #1); Kenneth Christopher Gavina (Reviewer #2)

Transaction Report:

DOI: <https://doi.org/10.1128/spectrum.02719-21>

February 6, 2022

Prof. Jian Li
Hubei University of Medicine
No.30, the people south road,Shiyan city, Hubei Province, China
Shiyan
China

Re: Spectrum02719-21 (A rapid and specific genotyping platform for Plasmodium falciparum chloroquine resistance via allele-specific PCR with lateral flow assay)

Dear Prof. Jian Li:

Link Not Available

Sincerely,

Tim Downing

Journals Department
Reviewer comments:

Reviewer #1 (Comments for the Author):

Major comments

1. L129 - 139: please present the results in plasmid copy number instead of concentration. There are other LOD report in other places in the manuscript (including L 184-189 in the discussion). A reported result in copy number would be more informative.
2. P. falciparum infection are often polyclonal. It would be interesting to add additional experiment with mix of the plasmids with and without the SNPs at different ratio.
3. L252 : precise which version of PlasmoDB was used. Also there is a reference to provide when using PlasmoDB. Please refer

to the website for proper citation

4. According to the manuscript, the PCR is performed separately and revealed using the lateral test. It needs to be more clear in the figure, as the current state could be misleading and imply that the full reaction is performed on the flow test.
5. Figures legends and labelling are insufficient. For example, all gels must present a detailed ladder (size).
6. How to you ensure that the system does not create cross contamination ?
7. Please assess the compatibility of your primers with *P. falciparum* isolates from different areas. For example, use the MalariaGen genomes to confirm your primers (one 3D7 strain is not enough for a test design).
8. The introduction and discussion would benefit from a better review of the relevant literature.

Minor comments

1. L52-54: rephrase "Despite of many simple, rapid, efficient, and high-throughput methods that have been developed, whereas Nested PCR with sequencing is still the 'gold standard' method for detecting SNPs and genotyping (7, 8)."
2. L55: be more precise "However, the test cycle is long, and the steps are too complicated, so the results could not be obtained in time.". What is the time frame? For which reason do we need a more rapid genotype profiling of malaria drug resistant parasites?
3. L61: "genotyping" do you mean genotype?
4. There are several typos and unprecise sentences in the manuscript. A full revision would be needed to make it clear.
5. L172 "too long a waiting time is still a problem to be solved"
6. L224 "the the"
7. L417: "genetype" => genotype
8. L 420 "respresent"
9. Make sure gene names are italicized
10. Figure 1 B is misleading by presenting a double strand DNA.
11. Figure 1C is unclear regarding the workflow.
12. The manuscript is understandable, but would require some additional work in terms of wording to be publication suitable.

Reviewer #2 (Comments for the Author):

Chloroquine (CQ) has been a primary treatment of malaria for decades however, CQ resistance continues to be an ongoing challenge in malaria elimination efforts, particularly in low-resource settings where the disease is endemic. In this study, Cheng and colleagues propose a PCR-LFA-based platform for both the detection and differentiation of wild-type and mutant *Pfcr* alleles as an alternative to traditional genotyping by sequencing. They validated this method using both plasmid surrogates and clinical specimens (dried filter blood spots). The PCR-LFA is a novel genotyping method that provides a cost-effective alternative to that of sanger sequencing, with rapid turn-around-time and similar levels of sensitivity and specificity. While Cheng and colleagues propose the utility of this platform for field studies and point-of-care, the requirements of nucleic acid extraction and PCR thermal-cycling ultimately limit this method's ability to truly be point-of-care.

Major Comments

1. Line 135/136 and Figure 3: The authors state that CVIET sensitivity is equivalent for both gel electrophoresis and LFA at 1.5pg/ μ l, the images on the lateral flow contradict this. While a band is clearly visible on the gel, no line can be seen on the lateral flow strip by the "naked eye".
2. Can the authors describe how they determined a LOD of 100 parasites/ μ L? (stated in Line 27, 161, 213 and 224). Were these values quantitated by microscopy? qPCR? Was a pro-bit analysis or dilution series performed to determine this value? No information is available in either the results of methods detailing this.
3. The authors propose this platform as an alternative for malaria genotyping at point-of-care and in the discussion reference other methods as inappropriate for use in the "field" due to the requirement of relying on laboratory instruments. While this is true, the same argument can be applied to the upstream DNA extraction and thermal-cycling required for the PCR-LFA method. Perhaps justifying this method as a cost-savings approach in low-resource or reference labs to replace sanger sequencing, or mention these caveats in the discussion, would be prudent.

Minor Comments

1. Spelling error in title, should read "allele-specific"
2. LFD acronym should be defined in Results as Methods is placed at end of manuscript
3. Have the authors considered or tested multiplexing the WT and mut primers and using a dual line lateral flow device? (i.e. possibly using combination of FitC, digoxin, and biotin) This would half the amount of reactions required to be run, as well as reduce the number of LFA cartridges used.
4. Can the authors describe what was used in the running buffer for the LFA (reagent added to sample pad after PCR products added)? Only listed information is pH. Further, can the authors comment as to whether non-specific binding/conjugation was observed if the LFA was allowed to sit beyond the 10 minutes?
5. Can the authors describe the instruments used for both PCR and sequencing?
6. Line 294: Can the authors the process by which 120 isolates were randomly selected. Although the authors reference their previous study, it may be helpful to briefly describe the specimens of the study as well.
7. Line 141-132: while this statement is theoretically true, unless the LFA assay is validated for utilizing 5 μ L of PCR product, you

cannot make this statement.

8. A couple of grammatical and spelling errors were observed throughout manuscript.

Staff Comments:

Preparing Revision Guidelines

Please return the manuscript within 60 days; if you cannot complete the modification within this time period, please contact me. If you do not wish to modify the manuscript and prefer to submit it to another journal, please notify me of your decision immediately so that the manuscript may be formally withdrawn from consideration by Microbiology Spectrum.

Dear Editor,

Cheng *et al*, present an innovative genotyping platform based on the selective detection of a PCR product using a lateral flow test.

The idea behind this assay is original and could contribute to malaria surveillance, by providing a low cost and simple to use test for SNP profiling in malaria clinical samples.

However, the manuscript would benefit from the following additions

Major comments

1. L129 – 139: please present the results in plasmid copy number instead of concentration. There are other LOD report in other places in the manuscript (including L 184-189 in the discussion). A reported result in copy number would be more informative.
2. *P. falciparum* infection are often polyclonal. It would be interesting to add additional experiment with mix of the plasmids with and without the SNPs at different ratio.
3. L252 : precise which version of PlasmoDB was used. Also there is a reference to provide when using PlasmoDB. Please refer to the website for proper citation
4. According to the manuscript, the PCR is performed separately and revealed using the lateral test. It needs to be more clear in the figure, as the current state could be misleading and imply that the full reaction is performed on the flow test.
5. Figures legends and labelling are insufficient. For example, all gels must present a detailed ladder (size).
6. How to you ensure that the system does not create cross contamination ?
7. Please assess the compatibility of your primers with *P. falciparum* isolates from different areas. For example, use the MalariaGen genomes to confirm your primers (one 3D7 strain is not enough for a test design).
8. The introduction and discussion would benefit from a better review of the relevant literature.

Minor comments

1. L52-54: rephrase “Despite of many simple, rapid, efficient, and high-throughput methods that have been developed, whereas Nested PCR with sequencing is still the ‘gold standard’ method for detecting SNPs and genotyping (7, 8).”
2. L55: be more precise “However, the test cycle is long, and the steps are too complicated, so the results could not be obtained in time.”. What is the time frame? For which reason do we need a more rapid genotype profiling of malaria drug resistant parasites?
3. L61: “genotyping” do you mean genotype?
4. There are several typos and unprecise sentences in the manuscript. A full revision would be needed to make it clear.
5. L172 “too long a waiting time is still a problem to be solved”
6. L224 “the the”
7. L417: “genetype” => genotype
8. L 420 “respresent”
9. Make sure gene names are italicized

10. Figure 1 B is misleading by presenting a double strand DNA.
11. Figure 1C is unclear regarding the workflow.
12. The manuscript is understandable, but would require some additional work in terms of wording to be publication suitable.

Replies to Editor and Reviewers,

First of all, we thank both reviewers and editor for your positive and constructive comments and suggestions.

Then, we have read the comments carefully and made correction accordingly. In this letter, we have provided a response to each comment below. In the revised manuscript, all changes were highlighted. Furthermore, the Tables, Figures and supporting information were modified based on the **Microbiology Spectrum style**. We tried our best to improve the manuscript and made some changes in the manuscript. These changes will not influence the content and framework of the paper. And here we remain the changing trace in revised paper with revisions mode.

Finally, the manuscript was edited for proper English language, grammar, punctuation, spelling, and overall style by Grammarly Premium and AJE Digital Editing.

Once again, we appreciate for Editors/Reviewers' warm work earnestly, and hope that the correction will meet with approval. The detail reply information for the editors and reviewers is as follows.

Reviewer comments:

Reviewer #1 (Comments for the Author):

Major comments

1. L129 - 139: please present the results in plasmid copy number instead of concentration. There are other LOD report in other places in the manuscript (including L184-189 in the discussion). A reported result in copy number would be more informative.

Answer: Thank you very much for your critical comments and suggestions. Based on your critical comments, we present the results in plasmid copy number instead of concentration and modify it in the manuscript and figure. We finally obtained the LOD of 3.38×10^5 copies/ μ l. The reason for the higher LOD could be the limitation of the methodology with fewer primers to choose from, a large number of AT sequences exist in the genome, and there is non-specific binding between primers and non-target sequences (Fig IA and IB). The presence of hairpin structures and primer dimers between the reverse primers, led to a lower amplification efficiency (Fig IC and ID). Further, the difficulty of amplification is doubled by the fact that *P. falciparum* is a particularly difficult organism to amplify and sequence, with an AT content of more than 80% in intergenic regions and introns, as well as a large number of AT-rich repeat sequences. In the next work, the use of digital PCR that can detect individual copy numbers is considered to further increase the LOD. In addition, when we detected SNPs by Recombinase Polymerase Amplification (RPA), we found that designing another universal forward primer before the forward primer can significantly increase the amplification efficiency of SNPs, and we can introduce this method into AS-PCR detection at a later stage. Please refer to line 27 on Page 2, lines 147-148, 153-157 on Pages 6-7, lines 248-255 on Page 10, and line 332-344 on Page 13, and refer to Fig 2, 3, 4, S1 and S2.

Figure I. Limitations of primers in methodology. (A) Nonspecific binding sites of forward primers; (B) Nonspecific binding sites of reverse primers; (C) Primer dimer in reverse primer; (D) Hairpin structure in the reverse primer.

2. *P. falciparum* infection are often polyclonal. It would be interesting to add additional experiment with mix of the plasmids with and without the SNPs at different ratio.

Answer: Thank you very much for your critical comments and suggestions. Based on your critical comments, we add additional experiments. Considering that LOD of CVIET was 3.38×10^5 copies/ μ l, amplification may fail after dilution again. After comprehensive consideration, the initial concentration of wild-type and mutant-type plasmids was 3.38×10^7 copies/ μ l, multiple infections were a mixture of wild-type and mutant-type in the ratio of 1:9 to 9:1, and the wild-type and mutant-type primers were used to amplify the mixed plasmids. According to the brightness of the

target bands amplified, the interference degree to genotyping under different proportions of mixed infection was analyzed. In the mixed plasmids of 1:9 and 9:1, the LOD of wild-type and mutant-type were 3.38×10^6 copies/ μ l. And added corresponding contents to the manuscript. Please refer to lines 162-166 on Page 7, lines 253-255 on Page 10, and Lines 338-344 on Page 13, and Fig S2.

3. L252 : precise which version of PlasmDB was used. Also there is a reference to provide when using PlasmDB. Please refer to the website for proper citation.

Answer: Thank you very much for your critical comments and suggestions. Based on your critical comments, we have updated and referenced the latest version of PlasmDB. Please refer to line 298 on page 11 and Reference 29.

4. According to the manuscript, the PCR is performed separately and revealed using the lateral test. It needs to be more clear in the figure, as the current state could be misleading and imply that the full reaction is performed on the flow test.

Answer: Thank you very much for your critical comments and suggestions. Based on your critical comments, we modified the detail, and explained that the whole reaction process is composed of two parts, LFA is carried out after AS-PCR. Please refer to lines 84-112 on pages 4-5, and lines 223-226 on Page 9.

5. Figures legends and labelling are insufficient. For example, all gels must present a detailed ladder (size).

Answer: Thank you very much for your critical comments and suggestions. Based on your critical comments, Markers were modified in Fig 2, 3, 4, S1, and S2, and the detailed ladder of Marker was modified in the legend. Please refer to lines 468-470, 474-478, and 482-484 on pages 17-18.

6. How to you ensure that the system does not create cross contamination ?

Answer: Thank you very much for your critical comments and suggestions. That's a good question. This same issue was taken into consideration before the experiments were carried out. For this project, contamination is most likely to occur in the PCR reaction. If the PCR process is well controlled, the problem will be solved. For PCR, the main contamination comes from the nucleic

acid. Thus, to ensure the accuracy of PCR test data and avoid cross-contamination due to aerosols, the PCR reaction and LFA assay were performed in a well-ventilated environment. In addition, to ensure the accuracy of data, nucleic acid removal solutions (FlaPurs Solution A and FlaPurs Solution B) were used to spray the whole experimental environment before and after experiments. According to the results of Figures 3A and 3B, the wild-type specific primers only amplified wild-type templates but did not amplify mutant templates. There was no presence of cross-contamination.

7. Please assess the compatibility of your primers with *P. falciparum* isolates from different areas. For example, use the MalariaGen genomes to confirm your primers (one 3D7 strain is not enough for a test design).

Answer: Thank you very much for your critical comments and suggestions. At the moment we only have clinical samples from various countries in Africa. We don't have any other strains on hand except the 3D7 strain. We tested more than 342 samples from our previously collected samples. I am confident of our results. Your opinion is very good and we will take it into full consideration in future study. I hope you understand the limitations of our present study.

8. The introduction and discussion would benefit from a better review of the relevant literature.

Answer: Thank you very much for your critical comments and suggestions. Based on your critical comments, we removed or modified unnecessary and incorrect statements, and replaced the appropriate references, references 7, 8, 10-15, 20, 23, 24, 26 and 29 were added and original references 9-12, 20, 22-28, 31, 32 and were deleted. In addition, the chronological order of references was also sorted. Please refer to lines 44-73 on pages 3-4, lines 184-274 on pages 7-11.

Minor comments

1. L52-54: rephrase "Despite of many simple, rapid, efficient, and high-throughput methods that have been developed, whereas Nested PCR with sequencing is still the 'gold standard' method for detecting SNPs and genotyping (7, 8)."

Answer: Thank you very much for your critical comments and suggestions. Based on your critical comments, we modified the detail. Please refer to lines 54-57 on page 3.

2. L55: be more precise "However, the test cycle is long, and the steps are too complicated, so the results could not be obtained in time". What is the time frame? For which reason do we need a more rapid genotype profiling of malaria drug resistant parasites?

Answer: Thank you very much for your critical comments and suggestions. Based on your critical comments, we modified the detail. These technologies often have testing cycles of more than a few days or more, and for patients with severe drug resistance problems, the best time to treat may be missed because of long waiting times. Please refer to lines 57-59 on page 3, and lines 201-202 on page 8.

3. L61: "genotyping" do you mean genotype?

Answer: Thank you very much for your critical comments and suggestions. Based on your critical comments, we modified the detail. Please refer to line 67 on page 3.

4. There are several typos and unprecise sentences in the manuscript. A full revision would be needed to make it clear.

Answer: Thank you very much for your critical comments and suggestions. Based on your critical comments, we have checked the manuscript carefully. The manuscript was edited for proper English language, grammar, punctuation, spelling, and overall style by Grammarly Premium and AJE Digital Editing. Please refer to lines 44, 52-59 on page 3 and so on.

5. L172 "too long a waiting time is still a problem to be solved"

Answer: Thank you very much for your critical comments and suggestions. Based on your critical comments, we modified the detail. Please refer to lines 200-201 on page 8.

6. L224 "the the"

Answer: Thank you very much for your critical comments and suggestions. Based on your critical comments, we modified the detail. Please refer to line 267 on page 10.

7. L417: "genotype" => genotype

Answer: Thank you very much for your critical comments and suggestions. Based on your critical

comments, we modified the detail. Please refer to line 455 on page 17.

8. L 420 "respresent"

Answer: Thank you very much for your critical comments and suggestions. Based on your critical comments, we modified the detail. Please refer to line 458 on page 17.

9. Make sure gene names are italicized

Answer: Thank you very much for your critical comments and suggestions. Based on your critical comments, we modified the detail. Please refer to line 11 on page 1.

10. Figure 1 B is misleading by presenting a double strand DNA.

Answer: Thank you very much for your critical comments and suggestions. Based on your critical comments, we modified the detail. Please refer to Fig 1B.

11. Figure 1C is unclear regarding the workflow.

Answer: Thank you very much for your critical comments and suggestions. Based on your critical comments, we modified the detail. Please refer to Fig 1C.

12. The manuscript is understandable, but would require some additional work in terms of wording to be publication suitable.

Answer: Thank you very much for your critical comments and suggestions. Based on your critical comments, we have checked the manuscript carefully. The manuscript was edited for proper English language, grammar, punctuation, spelling, and overall style by Grammarly Premium and AJE Digital Editing. Please refer to lines 44, 52-59 on page 3 and so on.

Once again, we appreciate your warm work earnestly and hope that the correction will meet with approval. All that you mentioned for us will significantly improve the quality of our manuscript. We thank you again for your positive and constructive comments and suggestions.

Reviewer #2 (Comments for the Author):

Chloroquine (CQ) has been a primary treatment of malaria for decades however, CQ resistance continues to be an ongoing challenge in malaria elimination efforts, particularly in low-resource settings where the disease is endemic. In this study, Cheng and colleagues propose a PCR-LFA-based platform for both the detection and differentiation of wild-type and mutant Pfcrt alleles as an alternative to traditional genotyping by sequencing. They validated this method using both plasmid surrogates and clinical specimens (dried filter blood spots). The PCR-LFA is a novel genotyping method that provides a cost-effective alternative to that of sanger sequencing, with rapid turn-around-time and similar levels of sensitivity and specificity. While Cheng and colleagues propose the utility of this platform for field studies and point-of-care, the requirements of nucleic acid extraction and PCR thermal-cycling ultimately limit this method's ability to truly be point-of-care.

Major Comments

1. Line 135/136 and Figure 3: The authors state that CVIET sensitivity is equivalent for both gel electrophoresis and LFA at 1.5 pg/ μ l, the images on the lateral flow contradict this. While a band is clearly visible on the gel, no line can be seen on the lateral flow strip by the "naked eye".

Answer: Thank you very much for your critical comments and suggestions. Based on your critical comments, we modified the detail. As requested by the reviewer 1, "Please present the results in plasmidcopy number instead of concentration". We convert the concentration to copy number. Then we considered changing the LOD of LFA in CVIET to 3.38×10^6 copies/ μ l (15 pg/ μ l). At 3.38×10^5 copies/ μ l (1.5 pg/ μ l) of CVIET, the weak bands were visible to the naked eye. There are several possible reasons for this. The first possibility is that LFA only added 2.0 μ l PCR product, while electrophoresis added 5.0 μ l. If the PCR products in the LFA are increased, the results can be the same as or better than the electrophoretic LOD results. It also reflected that the sensitivity of LFA was better than the electrophoretic results. The second possibility is that the camera equipment is not up to the high definition. The third possibility is that the combination of labeled colloidal gold is less effective than fluorescence and quantum dots. We can consider replacing the markers in the later stage. Please refer to lines 153-157 on pages 6-7, lines 248-255 on page 10, and Fig 3.

2. Can the authors describe how they determined a LOD of 100 parasites/ μ L? (stated in Line 27, 161, 213 and 224). Were these values quantitated by microscopy? qPCR? Was a pro-bit analysis or dilution series performed to determine this value? No information is available in either the results of methods detailing this.

Answer: Thank you very much for your critical comments and suggestions. Based on your critical comments, we modified the detail. These values were quantitated by microscopy. Parasitemia (parasites/l) was determined by counting the parasites during the erythrocytic stage against 200 leukocytes in the thick smears and multiplying by 8,000 as an estimated average total number of peripheral leukocytes for the individuals. Because this is a basic operation, so we did not mention the detail in the methods and only cited our previous literature (Reference 6 and 9). Furthermore, please refer to lines 382-392 on page 15. For the detailed information, please refer to subsection Collection of samples in Section Materials and Methods in reference 9 (Yao Y, Wu K, Xu M, Yang Y, Zhang Y, Yang W, Shang R, Du W, Tan H, Chen J, Lin M, Li J. 2018. Surveillance of Genetic Variations

Associated with Antimalarial Resistance of *Plasmodium falciparum* Isolates from Returned Migrant Workers in Wuhan, Central China. *Antimicrob Agents Chemother* 62.). The detailed information from reference 9 is as follows.

Blood samples (2 to 5 ml) were collected from patients with malaria in Wuhan Medical Treatment Center, Center for Disease Prevention and Control (Wuhan, China), and 14 hospitals in Wuhan from August 2011 to December 2016. Approximately 400 μ l of blood was spotted on Whatman 3MM filter paper, air dried and stored in an individually sealed polyethylene bag containing silica desiccant beads. The bags were stored at 20°C. These samples were subjected to One Step Malaria HRP2/pLDH (P.f/Pan) (Wondfo, Guangzhou, China) and Giemsa-stained thick and thin peripheral blood smear examination. Parasitemia (parasites/l) was determined by counting the parasites during the erythrocytic stage against 200 leukocytes in the thick smears and multiplying by 8,000 as an estimated average total number of peripheral leukocytes for the individuals. The identities of *Plasmodium* spp. were confirmed by real-time fluorescent quantitative PCR.

3. The authors propose this platform as an alternative for malaria genotyping at point-of-care and in the discussion reference other methods as inappropriate for use in the "field" due to the requirement of relying on laboratory instruments. While this is true, the same argument can be applied to the upstream DNA extraction and thermal-cycling required for the PCR-LFA method. Perhaps justifying this method as a cost-savings approach in low-resource or reference labs to replace sanger sequencing, or mention these caveats in the discussion, would be prudent.

Answer: Thank you very much for your critical comments and suggestions. Based on your critical comments, we changed the tone of the article and modified the detail, and explained that the whole reaction process is composed of two parts, LFA is carried out after AS-PCR. Although a thermocycler is required, the LFA replaces the time for Sanger sequencing and allows accurate genotyping in less than 2 hours. In the next plan, we are considering the use of a thermostatic amplification device instead of a thermal cycler, which can be completely detached from the laboratory instrumentation and carried out in the field. Please refer to lines 69-73 on pages 3-4, lines 216-219 on page 9.

Minor Comments

1. Spelling error in title, should read "allele-specific"

Answer: Thank you very much for your critical comments and suggestions. Based on your critical comments, we modified the detail. Please refer to line 2 on page 1.

2. LFD acronym should be defined in Results as Methods is placed at end of manuscript

Answer: Thank you very much for your critical comments and suggestions. Based on your critical comments, we modified the detail. Please refer to lines 86 on page 4, line 103 on page 5, and lines 217, 325 on page 12.

3. Have the authors considered or tested multiplexing the WT and mut primers and using a dual line lateral flow device? (i.e. possibly using combination of FitC, digoxin, and biotin) This would half the amount of reactions required to be run, as well as reduce the number of LFA cartridges used.

Answer: Thank you very much for your critical comments and suggestions. At the beginning of the experiment, we only considered saving the cost of antibodies. Based on your critical comments, we have also thought about this issue carefully. In the next step, CRISPR/Cas12a combined LFA was used to detect SNPs. We used two detection lines to detect wild-type and mutant at the same time, which was more cost saving.

4. Can the authors describe what was used in the running buffer for the LFA (reagent added to sample pad after PCR products added)? Only listed information is pH. Further, can the authors comment as to whether non-specific binding/conjugation was observed if the LFA was allowed to sit beyond the 10 minutes?

Answer: Thank you very much for your critical comments and suggestions. Based on your critical comments, we modified the detail. Buffer solution is 1 × PBS (calcium chloride 0.1 g/l, potassium chloride 0.2 g/l, potassium dihydrogen phosphate 0.2 g/l, magnesium chloride hexahydrate 0.1 g/l, sodium chloride 8 g/l, disodium hydrogen phosphate dodecahydrate 0.8975 g/l). In addition, the reaction in LFA shows true results within 10 minutes. After 10 minutes or more, non-specific binding may occur due to liquid volatilization and side flow, resulting in false positive results. Please refer to lines 326 on page 12.

5. Can the authors describe the instruments used for both PCR and sequencing?

Answer: Thank you very much for your critical comments and suggestions. Based on your critical comments, we modified the detail. Please refer to lines 294-295 on page 11.

6. Line 294: Can the authors the process by which 120 isolates were randomly selected. Although the authors reference their previous study, it may be helpful to briefly describe the specimens of the study as well.

Answer: Thank you very much for your critical comments and suggestions. Firstly, 120 samples were selected among those successfully sequenced for the pfcr gene to validate the performance of the AS-PCR-LFA detection system. These samples were all from the African region, mainly concentrated in West Africa and Central Africa. And these samples covered different Parasitemia (parasites/ μ l) ranges, including Low ($\leq 1,000$), Middle (1,001-9,999), and High ($\geq 10,000$).

7. Line 141-132: while this statement is theoretically true, unless the LFA assay is validated for utilizing 5 μ L of PCR product, you cannot make this statement.

Answer: Thank you very much for your critical comments and suggestions. Based on your critical comments, we modified the detail. Please refer to lines 166-168 on page 7, lines 248-255 on pages 9-10, and Fig 3.

8. A couple of grammatical and spelling errors were observed throughout manuscript.

Answer: Thank you very much for your critical comments and suggestions. Based on your critical comments, we have checked the manuscript carefully. The manuscript was edited for proper English language, grammar, punctuation, spelling, and overall style by Grammarly Premium and AJE Digital Editing. Please refer to lines 44, 52-59 on page 3 and so on.

Once again, we appreciate your warm work earnestly and hope that the correction will meet with approval. All that you mentioned for us will significantly improve the quality of our manuscript. We thank you again for your positive and constructive comments and suggestions.

March 25, 2022

Prof. Jian Li
Hubei University of Medicine
No.30, the people south road,Shiyan city, Hubei Province, China
Shiyan
China

Re: Spectrum02719-21R1 (A rapid and specific genotyping platform for Plasmodium falciparum chloroquine resistance via allele-specific PCR with lateral flow assay)

Dear Prof. Jian Li:

Your manuscript has been accepted, and I am forwarding it to the ASM Journals Department for publication. You will be notified when your proofs are ready to be viewed.

Sincerely,

Tim Downing
Editor, Microbiology Spectrum

Journals Department
Supplemental Material: Accept
Supplemental Dataset: Accept